# A Novel Upsampling and Context Convolution for Image Semantic Segmentation

**DOI:** 10.3390/s21062170

**Published:** 2021-03-20

**Authors:** Khwaja Monib Sediqi, Hyo Jong Lee

**Affiliations:** Division of Computer Science and Engineering, CAIIT, Jeonbuk National University, Jeonju 54896, Korea; kh.monib@gmail.com

**Keywords:** computer vision, convolutional neural networks, deep learning, pixel-wise classification, semantic segmentation

## Abstract

Semantic segmentation, which refers to pixel-wise classification of an image, is a fundamental topic in computer vision owing to its growing importance in the robot vision and autonomous driving sectors. It provides rich information about objects in the scene such as object boundary, category, and location. Recent methods for semantic segmentation often employ an encoder-decoder structure using deep convolutional neural networks. The encoder part extracts features of the image using several filters and pooling operations, whereas the decoder part gradually recovers the low-resolution feature maps of the encoder into a full input resolution feature map for pixel-wise prediction. However, the encoder-decoder variants for semantic segmentation suffer from severe spatial information loss, caused by pooling operations or stepwise convolutions, and does not consider the context in the scene. In this paper, we propose a novel dense upsampling convolution method based on a guided filter to effectively preserve the spatial information of the image in the network. We further propose a novel local context convolution method that not only covers larger-scale objects in the scene but covers them densely for precise object boundary delineation. Theoretical analyses and experimental results on several benchmark datasets verify the effectiveness of our method. Qualitatively, our approach delineates object boundaries at a level of accuracy that is beyond the current excellent methods. Quantitatively, we report a new record of 82.86% and 81.62% of pixel accuracy on ADE20K and Pascal-Context benchmark datasets, respectively. In comparison with the state-of-the-art methods, the proposed method offers promising improvements.

## 1. Introduction

Image semantic segmentation, which corresponds to pixel-wise classification of an image, is a vital topic in computer vision. It provides a comprehensive scenery description of the given image, including the information of object category, position, and shape. Semantic segmentation has an extensive array of applications ranging from scene understanding to self-driving cars and robot vision. Early methods that relied on hand-crafted feature extraction have been quickly superseded by deep learning technology [1]. The breakthrough of deep learning on various high-level computer vision tasks such as image classification [2,3] and object detection [4,5] has motivated computer vision scholars to explore the capabilities of such algorithms for pixel-level labeling problems such as semantic segmentation. The key advantage of deep learning approaches over the traditional one is their ability to learn rich representations for the problem at hand, i.e., automatic pixel labeling of an image in an end-to-end fashion instead of using manual feature extraction, which normally requires domain expertise and often too much fine-tuning to make them work in a particular scenario. Adapting convolutional neural networks (CNNs) for the task of semantic segmentation allows us to obtain rich details of object categories and scene semantics in an image.

Recent state-of-the-art methods employ an encoder-decoder structure for image semantic segmentation. The encoder part is a fully convolutional network (FCN) used to extract features at different resolutions. The decoder part, which is often termed as a “deconvolution”, is used to gradually upsample the feature maps obtained by the encoder into a semantically segmented output image. The FCN proposed by Shelhamer et al. [6] is arguably the first deep learning model designed for the task of image pixel-wise classification. The network is adapted from previous image classification networks such as AlexNet [7], VGGNet [8], and GoogLeNet [9]. The FCN replaces the last fully connected layers of image classification networks with convolutional layers to build an end-to-end trainable architecture for semantic segmentation. The network is able to take any arbitrary input image size and produce a predicted image with a resolution that corresponds to the size of input image. SegNet [10] uses FCN as an encoder and introduces a trainable decoder to gradually upsample feature map(s) obtained in the encoder part. Detailed object boundaries are recovered in the decoder path while the weights of the upsampling kernels are initialized using bilinear interpolation. SegNet achieves tremendous improvement in image semantic segmentation; however, the method is subject to severe loss of spatial information caused by pooling operation or convolution with stride and misses structure to utilize contextual semantics in the scene.

DeepLab [11] proposes atrous (also known as dilated) convolution and employs it in the last convolutional layers of FCN in an attempt to preserve spatial information in the network. The size of receptive field of view can be expanded using different dilation rates while the computational cost can be maintained constant as the dilation rates does not produces any overhead. DeepLabv2 [12], DeepLabv3 [13], and DeepLabv3+ [14] are extensions of DeepLab that employ atrous spatial pyramid pooling with conditional random field, parallel atrous convolution, and encoder-decoder with separable atrous convolutions, respectively. The DeepLab variants also poorly preserve spatial information of the image in the network. Figure 1a depicts the network structure of DeepLabv3+. As it can be seen, the spatial information in the image is lost in the first 4 convolution blocks. 

In this paper, we attempt to solve the problem of spatial information loss caused by pooling operations or convolutions with stride in a semantic segmentation network. In particular, we aim to solve this problem in the backbone of the network. We believe that the input high-resolution image contains rich fine-grained details that are crucial to be maintained in a semantic segmentation network. Hence, we propose a novel dense upsampling convolution (DUC) method based on guided filter to preserve the spatial information of the image in the network. The DUC upsamples the low-resolution feature map into a high-resolution feature map by propagating fine-grained details from the high-resolution image into the low-resolution feature map. Moreover, the semantic representations of the intermediate convolutional layer are concatenated with the output of the upsampling convolution feature map in order to produce denser high-resolution feature representation. We also address lack of structure to utilize representations in network. Benefiting from our proposed upsampling method, we further propose a novel dense local context (DLC) convolution method. We build the DLC module based on dilated convolution and it is able to effectively extract contextual information of objects in the scene. In comparison to spatial pyramid pooling proposed in DeepLab, our DLC can extract much denser contextual information and produces much larger receptive field of view to cover not only large-scale objects in the scene but also covers them densely.

In summary, the contributions of this paper are as follows:(1)We propose a novel upsampling convolution method to preserve the spatial information of the image in the network. The DUC is able to propagate fine-grained structure details from the input high-resolution image into the low-resolution feature map in an end-to-end trainable fashion.(2)To incorporate the object’s local contextual information into the network we develop a novel dense local context convolution method based on dilated convolution. The proposed method extracts dense contextual information using dilated convolutions in parallel and cascade with different dilation rates.(3)The proposed methods boost the performance of baseline networks for semantic segmentation in terms of pixel accuracy and mean intersection over union and outperform the state-of-the-art methods.

The remainder of this paper is organized as follows: Section 2 provides a comprehensive related literature review on semantic segmentation. The proposed method and theoretical analyses are described in Section 3. Section 4 presents the experimental setting and evaluation metrics with an empirical investigation on the effectiveness of the proposed method. In Section 5 we report qualitative and quantitative results and compares them with state-of-the-art methods. We present our conclusions in Section 6.

## 2. Related Works

Semantic segmentation is an active domain of research supplied by numerous challenging datasets [15,16,17,18]. Before the advent of deep learning technology, the best performing approaches relied heavily on manual extraction of features to classify pixels of an image independently. Generally, a patch of image is fed into a classifier, i.e., boosting [19], support vector machine [20], or random forest [21], to predict the probability of a class in center pixels. Improvements have been made by using richer information from local context [22] and structured prediction techniques [23,24]. However, the performance of these methods has always been compromised due to their limited expressive power of features. In the last few years, the deep learning technology that is used for image classification has been quickly transferred to the image semantic segmentation. Semantic segmentation includes both segmentation and classification, which raises the question of how to combine these two complicated tasks together.

The first family of deep learning-based approaches for semantic segmentation utilizes a cascade of bottom-up image segmentation, followed by deep-learning based region classification. For example, the bounding box proposals and masked regions suggested in [25] and [26] are used in [27] and [28] as inputs to a CNN for the pixel classification purposes. Similarly, Mostajabi et al. [29] rely on superpixel features for pixel-wise prediction. Even though these approaches delineate sharp boundaries delivered by a good segmentation, they cannot recover from any of its errors.

The second family of work relies on using convolutionally computed features for dense image labeling and joins them with segmentations that are obtained independently. Among the first, Farabet et al. [30] apply CNNs at multiple image resolution and then utilize a segmentation hierarchy to smooth the prediction results. Later, Krähenbühl and Koltun [23] proposed skip layers and concatenated the computed intermediate feature maps within the CNNs for pixel classification. Further Caesar et al. [31] proposed pooling of the intermediate feature maps using regional areas. These works still utilize segmentation algorithms that are disjointed from the CNN classifier’s results, thus risking commitment to a premature decision.

The third family of works use CNNs to directly provide dense category-level pixel labels, which makes it possible to even discard segmentation altogether [32]. The deep learning-based segmentation approaches directly apply CNNs to the whole image in a fully convolutional fashion by transforming the last fully connected layers of the previous image classification networks into a fully convolutional layer. In order to deal with object boundary delineation, SegNet uses an end-to-end trainable encoder-decoder structure where the encoder extracts feature maps at different resolutions and the decoder gradually upsamples the extracted features for pixel-wise prediction.

Recent work attempts to deal with the spatial localization problem and the aggregation of local context feature in the network. In the original encoder-decoder structure, several stages of pooling or convolution with stride in the encoder part reduces the spatial resolution for efficient computing performance. However, these operations eliminate the fine-grained structures such as object boundaries and edges in the image. For example, as shown in Figure 1a, DeepLabv3+ applies several dilated convolutions on the output of the last convolution layer containing very low fine-grained details of the original image. This information loss is caused by convolution with stride and pooling operations in the network. As a result, the spatial pyramid pooling performs poor multi-scale feature extraction on the image. In an attempt to solve the problem of the spatial information, Wu et al. [33] proposes a trainable guided filter in the network. As shown in Figure 1b, the guided filter jointly upsamples the output feature maps of the encoder by transferring structural details of the high-resolution input image into the low-resolution feature map of the last convolutional layer. In spite of improved segmentation performance, the guided filter layer bears extra computational cost in the network which makes it hard to train on large datasets. Similarly, Tian et al. [34] propose data-dependent upsampling approach to replace the traditional sequential decoder with a trainable data-dependent decoder. The data-dependent decoder works well in segmentation of classes with larger pixel distribution in the image. Their network fails to produce sharp delineation of object boundaries for small objects in the scene. The network structure of data-dependent upsampling is depicted in Figure 1c.

Our work is inspired by these networks. We extend them further by proposing a novel upsampling convolution and local context convolution method. In the upsampling convolution, we propagate the dense edges and saliency information of the high-resolution input image to the low-resolution feature map and further concatenate it with the feature representations coming from the intermediate layer in an end-to-end trainable fashion. In the context convolution, we extract dense contextual representations of objects in the scene using different dilated convolutions in parallel and cascade form.

## 3. Proposed Method

### 3.1. Joint Upsampling 

Before we introduce our proposed DUC method, we revisit the joint upsampling procedure. Among many popular upsampling filters, guided filter [35] stands out the crowd owing to its simplicity, robustness, and fast speed. A guided filter is an edge-preserving smoothing operator that can produce an output image while transferring structural details of the input image itself or a different image. Given a high-resolution guided image $I_{h}$ and a low-resolution target image $I_{l}$, guided joint upsampling aims to generate a high-resolution output image $O_{h}$ by transferring structural details from guided image $I_{h}$. Assuming that $O_{h}$ is a linear transform of $I_{h}$ in a square window $w_{k}$ centered at the pixel k, then it can be formally defined as:(1)$$O_{h}^{i}=a_{k}^{i}I_{h}^{i}+\;b_{k}^{i},\;\forall_{i}\;\in w_{k}$$
where $\left(a_{k},b_{k}\right)$ are assumed to be constant coefficients in the local square windows, $w_{k}$, and i indicates the $i^{th}$ pixel of the image. The linear model ensures that the high-resolution output image has an edge only if the guidance image has an edge, because $\nabla O_{h}^{i}=a_{k}^{i}\nabla I_{h}^{i}$. 

To specify the model coefficients $\left(a_{k},b_{k}\right)$, a constant from $I_{l}$ is required. The model outputs $O_{h}$ by subtracting some unwanted components from $I_{l}$, such as noise or texture:(2)$$O_{h}^{i}=I_{l}^{i}-\;n_{i}$$

A cost function is used to minimize the difference between $O_{h}$ and $I_{l}$, while preserving the linear model in Equation (1). After computing $\left(a_{k},b_{k}\right)$ for all local square windows, $w_{k}$, in the image, the high-resolution output image $O_{h}$ is computed as:(3)$$O_{h}={\bar{a}}_{i}\times \;I_{h}^{i}+\;{\bar{b}}_{i}$$
where × denotes element-wise multiplication and (${\bar{a}}_{i}$,${\bar{b}}_{i}$) are the coefficients averaged over all windows overlapping i.

The actual guided filter is used as a post-processing operation. It is not differentiable and thus cannot be trained in an end-to-end manner with the FCNs. To boost the performance of the FCN for upsampling, Xu et al. [36] propose edge-aware filters by transforming the simple guided filter into a learnable layer, which enables both the guided filter layer and the FCNs to be trained simultaneously by providing direct guidance from the high-resolution image. We build our dense upsampling method based on this work and then employ a DLC module to incorporate rich semantic details to improve the performance of semantic segmentation network. 

### 3.2. Dense Upsampling Convolution (DUC) 

We propose a novel DUC method based on a guided filter to preserve the spatial information of the image in the network as shown in Figure 2. The DUC upsamples a low-resolution feature map to a high-resolution feature map by transferring spatial details of high-resolution image into the low-resolution feature map. As the computational graph of the DUC method depicted in Figure 3 shows, a transformation function of g(I) is used to generate the guidance maps of $G_{h}$ from the high-resolution input image. The transformation function g(I) includes a two-layer pointwise $\left(1\times 1\right)$ convolutional block consisting of a normalization layer in between and a rectified linear unit (ReLU) activation function. The guidance maps are lightweight image representations that transfer the object boundary and edge information. We employ a convolution layer with stride 4 to process the $G_{h}$ to produce an output equivalent to the size of ${\bar{G}}_{h}$, where the spatial size of ${\bar{G}}_{h}$ corresponds to the size of the intermediate feature map $I_{m}$. A dilated convolution is employed on the low-resolution feature map $I_{l}$ to extract feature representations of ${\bar{I}}_{l}$. The guided representations of $G_{h}$ and feature representations of ${\bar{I}}_{l}$ are fed into a pointwise convolution. Following the Equation (3), the pointwise convolution produces coefficients of ${\bar{a}}_{i}$ and ${\bar{b}}_{i}$. We choose the pointwise convolution owing to its robust feature extraction capability and the reduction of parameters in the network. It is also possible to use any other standard convolution other than the pointwise convolution. In the last stage, a bilinear interpolation is used to upsample the obtained guidance map with the low-resolution feature map, thereby yielding an output size that commensurate to the intermediate layer features of $I_{m}$.

In order to propagate denser spatial details into the network, we further concatenate the output of the bilinear interpolation with the feature maps flowing from the intermediate layer ($I_{m})$. Consequently, the final output feature map attains a size that is equivalent to the size of feature map in the C3 (Convolutional layer 3) of the model. In Figure 2 we show the complete implementation of DUC module in the baseline network. Notably, the DUC module is trainable in an end-to-end fashion, thus it can learn features from scratch. The proposed DUC solves two closely related problems. (1) It solves the problem of spatial information loss, which is caused by pooling or convolution with stride, by transferring fine-grained details from the high-resolution input image into the low-resolution feature map of the last convolution layer. (2) It recovers the missing salient information regarding the object’s boundary by concatenating feature representations from the intermediate layer. The proposed DUC can be implemented with any CNN network. In our experiments, we employ DUC with ResNet [3] using different depths of [52,101,152, and 269] and report the results.

### 3.3. Dense Local Context (DLC) Convolution

Objects in the scene prevail in small or large scales, making it difficult to extract the proper feature representation needed for semantic segmentation. To overcome this problem, DeepLab proposes an atrous spatial pyramid pooling (ASPP) module and applies it using different dilation rates on the output convolutional layer of the encoder and fuse the final output to attain multi-scale feature representations. Atrous convolution can exponentially enlarge the receptive field (RF) of a convolution kernel. Let *d* and *k* denote the dilation rate and kernel size of the atrous convolution layer, respectively. Then the equivalent RF size of the kernel is obtained as proposed in [11] as Equation (4).
(4)$$RF=\left(d-1\right)\;\times \;\left(k-1\right)+k$$

DeepLab employs ASPP in parallel using dilation rates d= 6, 12, 18, and 24. However, ASSP implementation is associated with the following issues. Firstly, it is not dense enough to capture features of large-scale objects in difficult scenes. Secondly, as shown in Figure 1a, ASPP is employed after the last layer of the encoder which produces a low-resolution feature map (1/16 size of the original image). At this stage, the spatial information of the image is lost by a factor of 16, thus ASPP fails to extract rich feature of the image. For example, assuming an image with a size of 512×512, height and width, respectively, ASSP is applied to the feature map with a reduced spatial resolution size of 32×32. This produces a poor multi-scale feature extraction. Thirdly, the implementation of ASPP with a dilation rate (d) of 24 is ineffective for low resolution images. Based on Equation (3), ASPP with the dilation rate of 24 enlarges the RF size to 49, which is larger than the feature map size obtained at the last convolutional layer (feature map = 32). Partially, we solve this problem with the previously proposed DUC method. The DCU produces relatively higher resolution feature by upsampling the output feature map of the last convolutional layer with the fine-grained details from the high-resolution input image.

We further employ a DLC module based on DenseASPP [37] to replace the ASSP module of the DeepLabv3. In a closely related line of research to our work, Ding et al. [38,39] propose context contrasted local (CCL) model as an alternative to ASPP for multi-level feature extraction. Notably, our method is not only different from the approaches used in CCL but also outperforms it (ref. Section 5). As an example, CCL uses a combination of local (delicate) convolution and an atrous (coarse) convolution in parallel to extract multi-scale feature map in the network. In contrast, DLC combines the benefits of parallel and cascade atrous convolutions to produce larger RF and achieves denser multi-scale features of objects in the scene. Suppose there are two convolutional layers with the filter size of $K_{1}$ and $K_{2}$. Then the new RF can be achieved from the stack of these two convolution layers as Equation (5).
(5)$$K=\;K_{1}+K_{2}-1$$

Following Equations (4) and (5) CCL with the atrous convolution rate of $\left[d=3,\;6,\;12,\;\mathrm{and}\;18\right],$ and four local convolutions of size $\left(3\;\times \;3\right)$ produces a relatively small receptive field of view (i.e., RF = 49), whereas DLC with the same atrous convolution rate of $\left[d=3,\;\;6,\;\;12,\;\mathrm{and}\;18\right]$ assembles a much larger receptive filed view (i.e., RF = 79). This enables DLC to cover not only larger objects in the scene but also covers them densely for better segmentation results. Figure 4 presents the architecture of the proposed DLC.

The upsampled feature map obtained from the DUC is fed into a 1×1 convolution layer to reduce the number of parameters in the network. Thereafter, the feature maps of size “1/8×” are convolved with several atrous convolutions using dilation rates of 3, 6, 12, and 18, respectively. The output of each dilated convolution is further concatenated with the input of the next dilated convolution with a larger dilation rate. Compared to DenseASPP, we use a 1×1 convolution before the input of each dilated convolution, resulting in a less complex network while extracting richer feature of the image. Further, we omit the dilation rate of 24 in our DUC module. This brings us two benefits: (1) we achieve a less complex network with fewer parameters, and (2) we attain an RF size that is big enough to cover large-scale objects. The DLC module produces about 1.5M parameters, which is only 23% of DenseASPP (nearly 6.48M). Figure 4 shows the detailed implementation of DLC in the network.

## 4. Experiments

In this section, we first introduce the benchmark datasets used in this experiment. We choose ResNet [3] as the backbone of our network. Finally, we evaluate the performance of the proposed method using standard evaluation metrics for pixel-wise classification and report the results. 

### 4.1. Dataset

We verify the effectiveness of the proposed methods on three challenging datasets: ADE20K [18], Pascal-Context [5] and Cityscapes [15]. ADE20K is a densely annotated dataset for semantic segmentation. It contains diverse annotations of scenes, objects, parts of objects, and in some cases even parts of parts objects. ADE20K contains 20,210 images in the training set, 2000 images in the validation set, and 3000 images in the testing set. Of the total 3169 annotated class labels, 2693 are object and stuff classes and 476 are classes belonging to the part of the objects. Figure 5 shows image and label samples from ADE20K dataset. 

For each object, there is additional information about whether it is occluded or cropped and other attributes. The images in the validation set are exhaustively annotated with parts, while the part annotations are not exhaustive over the images in the training set. On average there are 19.5 instances and 10.5 object classes per image. Pascal-Context dataset is a pixel-wise annotated extension of the PASCAL VOC detection challenge. The total 10,103 scene images are comprised of 4998 and 5105 images for training and validation set, respectively. We followed the standards metric provided in [5] by using all 59 class labels, including one background class to evaluate the performance of the network. Cityscapes dataset, which is designed for semantic urban scene understanding, has 5000 high-quality fine pixel-level annotations. The images are divided into three splits of numbers 2975, 500, and 1525 for training, validation, and testing, respectively. Besides, 20,000 coarsely annotated images are provided for two settings in comparison, i.e., training with only fine data or with both fine and coarse data.

### 4.2. Implementation Details 

We use PyTorch [40], an open-source deep learning framework, to implement our network architecture. Initially, we implement our method using ResNet50 [3] in the backbone. “Poly” learning policy [41], which defines current learning rate equals to the base learning rate multiplied to ${\left(1-\frac{iter}{\max\_iter}\right)}^{p}$, is used as a learning update strategy. The initial learning rate is set to 0.001 and *p* is set to 0.9. We use data augmentation techniques of random vertical flipping and random scaling from 0.5 to 2. We also benefit from soft computing preprocessing based on fuzzy technique [42,43,44,45] to avoid peculiarities in the images. The images are then cropped to 480 × 480 and fed to the network. The network is trained using 120 and 80 epochs for ADE20K and Pascal-Context datasets, respectively. Stochastic gradient descent (SGD) is used as an optimizer, and the momentum is set to 0.9 with weights decay value of 1e−4. All experiments are conducted on 4-TitanX GPUs (12 GB of memory per GPU) in parallel, where the loss is computed from multiple GPUs simultaneously. We also investigate the impact of our method on other ResNet variants with deeper layers that are designed for image semantic segmentation by keeping the default configuration settings of each network. 

### 4.3. Loss Function 

Optimization of a deep learning model is driven by loss function. In order to minimize the overall loss, the parameters of the neural network are updated by backpropagation method. We use the standard multi-class cross-entropy loss, also called logarithmic loss, as commonly used in multi-classification models. Cross-entropy loss decreases as the predicted probability converges to the actual label. Cross-entropy in semantic segmentation is defined as: (6)$$L=-\;\sum_{n=1}^{N}\sum_{k=1}^{K}\sum_{j=1}^{P}t_{nk}\log P$$
where N, K, and P indicate the batch-size, the number of classes, and the predicted pixels, respectively, and t represents a one-hot target vector and $t_{k}=1$ when k is a true label. 

### 4.4. Performance Evaluation

Standard evaluation metrics are used to assess the performance of the semantic segmentation algorithms. These criteria are the variation of pixel accuracy (PA) and intersection over the union. Let k+1 denote the number of classes from $L_{0}$ to $L_{k}$, including a background or void class, and $p_{ij}$ is the number of pixels of class i that are inferred to belong to class j. In other words, $p_{ii}$ denotes the number of true positives, while $p_{ij}$ and $p_{ji}$ are often referred to as false positives and false negatives, respectively, although either of them can be the sum of both false positives and false negatives. *PA* has often been adopted to measure the performance of semantic segmentation algorithms. It is simply the computation of the ratio between the amount of properly classified pixels and the total number of pixels; mathematically it is represented as follows:(7)$$PA=\;\frac{\sum_{i=0}^{k}p_{ii}}{\sum_{i=0}^{k}\sum_{j=0}^{k}p_{ij}}$$

Mean PA (*mPA*) is a slightly improved PA metric, in which the ratio of correct pixels is computed on a per-class (per-category) basis and then averaged over the total number of classes. *mPA* is defined as:(8)$$mPA=\;\frac{1}{k+1}\overset{k}{\underset{i=0}{\sum }}\frac{p_{ii}}{\sum_{j=0}^{k}p_{ij}}$$

Mean intersection over union (mIoU)—originally proposed in the pascal visual object classes challenge [46]—is another standard metric used in semantic segmentation. It is the ratio of intersection over the union of the predicted segmentation and the ground truth. The ratio can be reformulated into the number of true positives (intersection) over the sum of true positives, false positives, and false negatives (union). The IoU is computed on a per-class basis and then averaged over the total number of classes, which is referred to as mIoU. Formally, it is denoted as:(9)$$mIoU=\;\frac{1}{k+1}\overset{k}{\underset{i=0}{\sum }}\;\;\frac{p_{ii}}{\sum_{j=0}^{k}p_{ij}+\sum_{j=0}^{k}p_{ji\;}+\;p_{ii}}$$

Among all these metrics, the mIoU stands out of the crowd as it is a widely used criterion due to its representativeness and simplicity.

## 5. Results

We report qualitative and quantitative analyses of our method on benchmark datasets, described in Section 4. We compare the results of our method with those of the state-of-the-art. 

### 5.1. Qualitative Results

We report visual results on the validation set of ADE20K and Pascal-Context datasets for image semantic segmentation. Our method presents a good ability in finding missing parts of small- and large-scale objects in complex scenes, as demonstrated by the ADE20K scene parsing dataset. Figure 6 shows the robustness of our method when delineating a small-scale object in the scene. 

As can be seen in the first and second rows, our method treats objects boundary far better than the baseline model does. Looking at the row three, the baseline architecture misclassifies “pillows” as “bed”. Our method corrects this error by classifying “pillows” as “pillows” on the bed. This is attributed to the robust capability of the DUC module in the network. The DUC is designed to preserve the sharp boundary of objects in the scene, which generates better segmentation output. The excellent performance of the network is notable when it comes to the utilization of the context in the scene. It can be seen that the proposed method has excellent ability in delineating edges of the objects in the scene—much better than the baseline network.

Moreover, we can see that our model has semantically well classified small- and large-scale objects in the scene in comparison to the baseline model. This shows the superiority of the proposed method. Visual examples of Pascal-Context classification are shown in Figure 7. In row one of the Figure 7, the baseline model treats the “rocks” as part of the “mountain”, whereas our method corrects this error and classifies it as “rock”. Following that, for “bus” and “animal” our method more precisely outlines the object boundary than the baseline algorithm. The great effect of the proposed DLC module becomes clear in the fourth row. DLC utilizes contextual information to properly distinguish the “person” from the “tree”. 

### 5.2. Quantitative Results

In this research, we aggressively evaluate the effectiveness of the proposed method with regard to three large-scale benchmark datasets described in Section 4.1. We test our method using deeper neural networks with single- and multi-scale entries. A single-scale input refers to the original image size, whereas multi-scale is multiple resized input images. In a multi-scale entry, the features are shared in the network and then merged for pixel-wise prediction. Deeper networks are proven to achieve robust performance on large-scale data classification. Hence, we conduct experiments using different depths of ResNet in the backbone. We test pre-trained ResNet with a depth of 50, 101, 152, and 269.

It is evident that keeping the default setting of the network and increasing the depth from 50 layers to 269 layers improves the final score (average of mIoU and PA) from 62.93% to 65.49%, with 2.56% absolute improvement in multi-scale input setting. Detailed final scores of our method with different ResNet depth in the backbone are listed in Table 1.

A comparison with state-of-the-art methods on the ADE20K dataset is given in Table 2. We report new records of performance on ADE20K dataset by achieving 46.41% of mean IoU and 82.86% of PA using ResNet-101 in the backbone of our network architecture. The proposed method outperforms state-of-the-art methods on pixel accuracy. Further, we achieve the highest mean IoU and pixel accuracy over the validation set of the Pascal-Context dataset. Table 3 shows the comparison of our method with state-of-the-arts.

Further, we achieve the highest mean IoU and pixel accuracy over the validation set of the Pascal-Context dataset. Table 3 shows the comparison of our method with state-of-the-arts. Following the standard evaluation metric in [5], we consider all 59 classes, plus one background class for the evaluation and report the results. Our method produces an encouraging score in comparison with previous best available methods by achieving 56.1% and 81.62% of mIoU and pixel accuracy, respectively.

Finally, we investigate the effectiveness of our method on the new publicly available dataset of Cityscapes. Our method remarkably outperforms the state-of-the-arts on the test split of the Cityscapes dataset by achieving a mIoU of 83.3%. Table 4 shows the detailed comparison on Cityscapes dataset. 

## 6. Conclusions

In this study, we have addressed the problem of spatial information loss and missing contextual details for image semantic segmentation using deep learning. We propose a dense upsampling convolution method based on guided filtering that is able to effectively preserve the spatial details in the network by transferring fine-grained structures from the input high-resolution image to the low-resolution feature map in an end-to-end trainable fashion. We further propose a dense multi-scale context convolution module based on atrous convolution that is able to incorporate rich local context description in the network. We tested the impact of the proposed method on ADE20K, Pascal-Context and Cityscapes benchmark datasets. Visual result revealed that the proposed method classifies object boundaries at a higher accuracy than that of the recent competitive models, which demonstrates the effectiveness of our method. We also included single- and multi-scale inputs in our experiments to find their correlation with respect to the problem of pixel-wise prediction. The experimental results showed that multi-scale inputs promise better performance than the single-scale entry. We also, studied the impact of deeper ResNet in the backbone with regard to the performance of semantic segmentation output. The results indicated that the depth of the backbone network is directly proportional to the performance of semantic segmentation (i.e., the deeper the network, the better the performance). Despite the success of this approach, in future work, we aim to improve the prediction accuracy for “parts” and “parts of parts” of the objects in the scene as provided by the ADE20K dataset. Adopting this approach to object detection and localization is another excellent domain of research to invest on. 

## Figures and Tables

**Figure 1 sensors-21-02170-f001:**
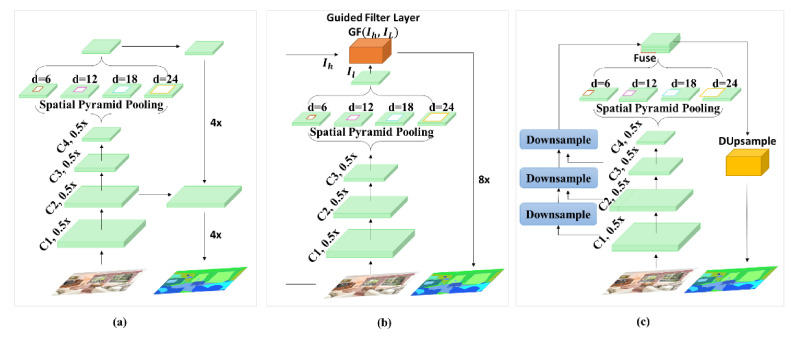
Overview of DeepLabv3 network architecture (**a**), implementation of guided filter layer over DeepLabv2 (**b**), and illustration of DUpsampling module employed after the last layer of DeepLabv3 (**c**).

**Figure 2 sensors-21-02170-f002:**
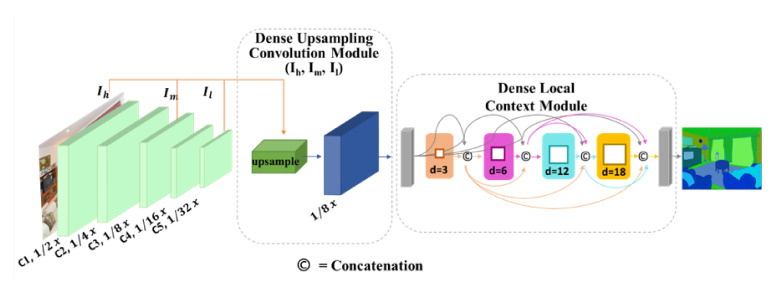
Network overview: The DUC is implemented after the last convolutional layer of the backbone network to upsample the low-resolution feature map into a high-resolution feature map by transferring structural details from the high-resolution guidance image. The DLC is used on the output of the DUC layer in order to extract dense multi-scale local contextual representations. Best viewed in color.

**Figure 3 sensors-21-02170-f003:**
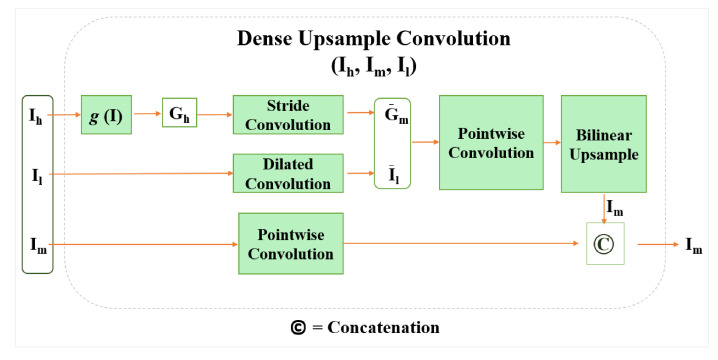
Computational graph of the proposed dense upsampling convolution (DUC) method.

**Figure 4 sensors-21-02170-f004:**
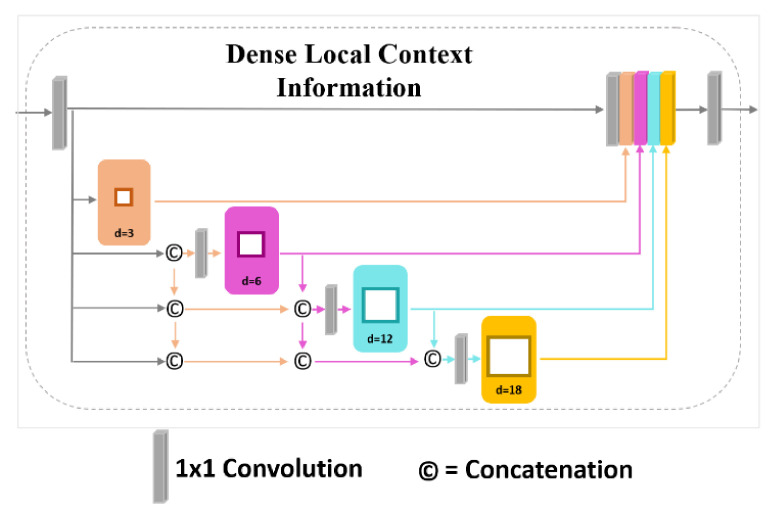
Detailed overview of Dense Local Context (DLC) module.

**Figure 5 sensors-21-02170-f005:**
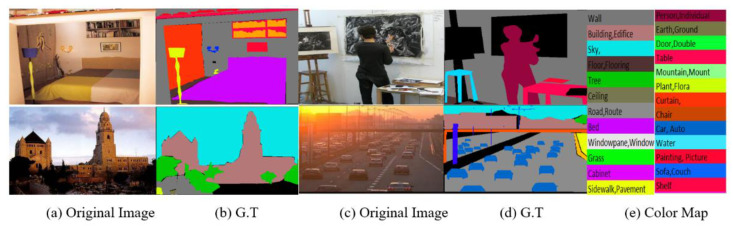
Image samples from ADEK 20K dataset. G.T stands for Ground Truth.

**Figure 6 sensors-21-02170-f006:**
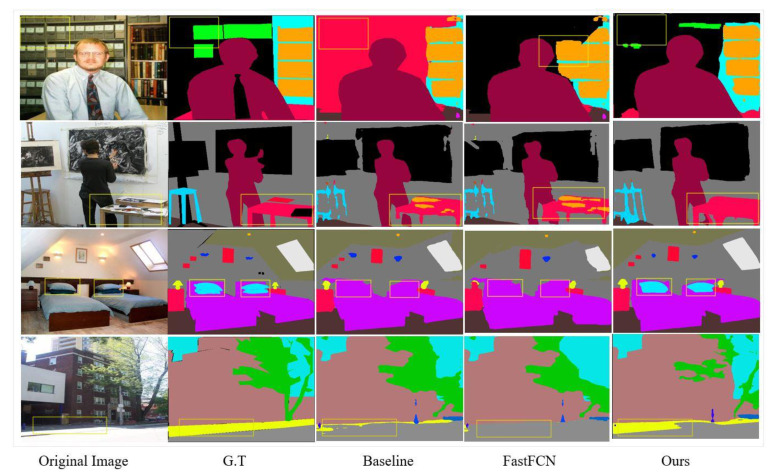
Visual improvements on ADE20k scene parsing dataset. Our method corrects the error of the baseline model and produces more accurate and detailed results.

**Figure 7 sensors-21-02170-f007:**
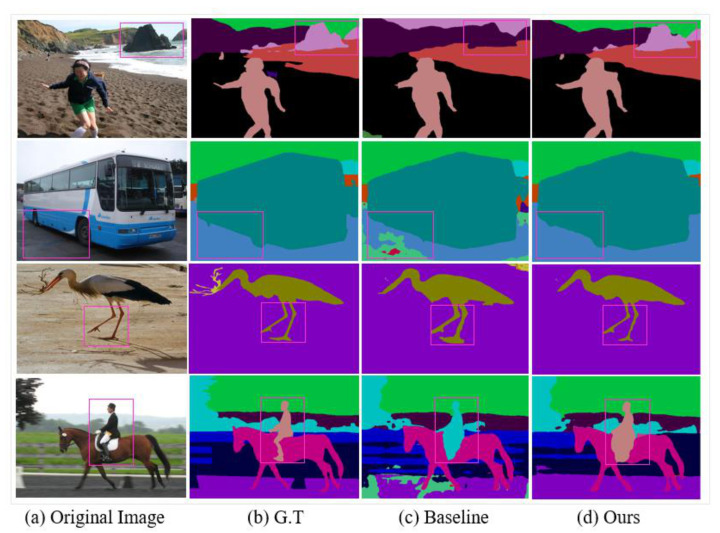
Visual improvements on Pascal-Context dataset. Our method well considers boundary information. Consequently, produces more accurate and detailed results.

**Table 1 sensors-21-02170-t001:** Deeper pre-trained ResNet attains higher performance. Numbers in the parentheses refer to the depth of ResNet. SS and MS denote single- and multi-scale testing, respectively. Experiments are conducted on the ADE20K dataset.

Method	Mean IoU (%)	Pixel Acc. (%)	Final Score
Ours (50) + SS	43.82	81.23	62.53
Ours (101) + SS	44.21	82.71	63.46
Ours (152) + SS	44.86	82.91	63.89
Ours (269) + SS	46.26	83.11	64.69
Ours (50) + MS	44.16	81.70	62.93
Ours (101) + MS	46.41	82.86	64.64
Ours (152) + MS	46.88	83.66	65.27
Ours (269) + MS	47.16	83.82	65.49

**Table 2 sensors-21-02170-t002:** Comparison with state-of-the-art methods on ADE20K dataset. Our method outperforms the state-of-the-arts on pixel accuracy.

Method	Mean IoU (%)	Pixel Acc. (%)	Final Score
SegNet [10]	21.64	71.00	46.32
DilatedNet [47]	32.31	73.55	52.93
CascadeNet [48]	34.90	74.52	54.71
RefineNet [49]	40.70	-	-
PSPNet [50]	43.29	81.39	62.34
FastFCN [51]	44.34	80.99	62.67
EncNet [22]	44.65	81.69	63.17
CPNet [52]	45.39	81.04	63.21
CGBNet [37]	44.90	82.10	63.50
ResNeSt [53]	**46.91**	82.07	64.49
Ours	46.41	**82.86**	**64.64**

**Table 3 sensors-21-02170-t003:** Comparison with state-of-the-art methods on Pascal-Context dataset. Our method outperforms the state-of-the-arts.

Method	Mean IoU (%)	Pixel Acc. (%)	Final Score
DeepLabV2 [12]	45.70	-	-
RefineNet [49]	47.30	-	-
PSPNet [50]	47.80	-	-
EncNet [22]	51.70	-	-
Dupsampling [34]	52.50	-	-
DANet [54]	52.60	-	-
FastFCN [51]	53.10	79.12	66.11
CPNet [52]	53.90		
CGBNet [37]	53.40	79.60	66.50
DRAN [55]	55.40	79.60	67.50
Ours	**56.10**	**81.62**	**68.86**

**Table 4 sensors-21-02170-t004:** Comparison with state-of-the-art methods on Cityscapes dataset. Out method outperforms the state-of-the-arts.

Method	Mean IoU (%)
DilatedNet [47]	66.8
DeepLabV2 [12]	70.4
RefineNet [49]	73.6
PSPNet [50]	78.4
DenseASPP [36]	80.6
CGBNet [37]	81.2
DRAN [55]	82.9
Ours	**83.3**

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
