# Peer review of "A Novel Upsampling and Context Convolution for Image Semantic Segmentation"

_sensors, 2021, doi:10.3390/s21062170_

Round 1
Reviewer 1 Report
This manuscript proposed a dense upsampling convolution method and a dense local context convolution method to overcome spatial details loss and combine local context information. The matter is of interest. In addition, the idea of the article is clear and the experimental results are illustrative. The authors should take care of the layout of this article. The problem of format inconsistencies is serious (such as Line 210-218). However, the paper suffers the following limits, authors should carefully revise this manuscript before its publication:
My biggest concern is the author's description of the innovation of the article is not very prominent, and too much space is used to explain common-sense knowledge.
Besides, it might be better if the author could add some visual comparison with other algorithms in the experiment of Qualitative Results.
In Section 5.2, the results of “Our” in Table 1 and 2 seems to be some gap. Assuming ResNet-101 is used in Table 2, its results outperformed that in Table 1 with ResNet-101.
Author Response
Thank you for your valuable comments. We answered to your comments in a separated file.

Reviewer 2 Report
The paper is very interesting, well written and well structured. The methodological approach is based on a more than satisfactory scientific rigor.
The abstract briefly describes the most important qualitative and quantitative results achieved, offering an exhaustive overview of all the work carried out.
1. Please improve the quality of Figure 1 as it is difficult to view.
2. The proposed methodology is scientifically convincing, well structured and, especially, well described. All the mathematical formulas have been commented in sufficient detail. However, it is noted that some of them are not original, so I believe that each of them must be associated with a relevant bibliographic reference.
3. The processed images, however, may be affected by uncertainty and / or inaccuracy. Therefore, in these cases, it is necessary to carry out a pre-processing of the images that takes these peculiarities into account. Then soft computing pre-processing techniques based on fuzzy and / or neuro fuzzy techniques should be used for these objectives.
4. I advise the Authors to insert in the paper some comment sentences for images that need pre-processing of this type by inserting the following relevant works in the bibliography:
Versaci, M., Morabito, F. C., & Angiulli, G. (2017). Adaptive image contrast enhancement by computing distances into a 4-dimensional fuzzy unit hypercube. IEEE Access, 5, 26922-26931. doi: 10.1109/ACCESS.2017.2776349
Jeon, G., Anisetti, M., Damiani, E., & Monga, O. (2018). Real-time image processing systems using fuzzy and rough sets techniques. doi:10.1007/s00500-017-2999-3
Rahim, S. S., Jayne, C., Palade, V., & Shuttleworth, J. (2016). Automatic detection of microaneurysms in colour fundus images for diabetic retinopathy screening. Neural computing and applications, 27(5), 1149-1164. doi: 10.1007/s40708-016-0045-3
Orujov, F., Maskeliūnas, R., Damaševičius, R., & Wei, W. (2020). Fuzzy based image edge detection algorithm for blood vessel detection in retinal images. Applied Soft Computing, 94, 106452. doi:/10.1016/j.asoc.2020.106452
5. Also, please specify how was (9) obtained.
Author Response

(The authors gave the same response as above.)
